# Determinants of Economical High School Students' Attitudes toward Mobile Devices Use

**Mihaela Moca * and Alina Badulescu ***

Department of Economics and Business, University of Oradea, 410087 Oradea, Romania
* Correspondence: moca.mihaela@gmail.com (M.M.); abadulescu@uoradea.ro (A.B.)

**Abstract:** Due to recent considerable technology breakthroughs in the education sector, new tools have been developed to improve learning. Motivating students to use new devices for learning rather than just for amusement, however, is a difficulty. The COVID-19 pandemic prompted the adoption of technological devices for course delivery, thereby highlighting the significance of mobile learning (m-learning) and allowing educators, students, and other stakeholders in the education sector to recognize its potential, advantages, drawbacks, and challenges. As m-learning has been an essential aspect of education for some time now, there is growing interest in assessing its long-term viability and usefulness across various educational domains, including economics. New technologies like computers, the internet, and related tools can help by bringing life to the classroom, gauging student progress, simulating economic activities and phenomena, and teaching vital skills needed for the economic world, like entrepreneurship. This study aims to explore the potential of incorporating new technologies in economic education, we study the tendency of the economical high school students towards using mobile devices for learning activities. A total of 407 participants were involved in research, the data from these respondents being collected with the help of a questionnaire survey. The original technology acceptance model (TAM) has been extended and the role of various external factors such as the subjective norm, learning autonomy, facilitating conditions or self-efficacy has been addressed. A list of hypotheses was proposed to validate the underlying model and provide guidance on how external factors affect attitudes towards using mobile devices. The empirical results indicated that perceived ease of use and perceived usefulness are significant predictors to explain the attitudes towards mobile devices use and m-learning and the analyzed external factors have a positive influence on them. In terms of methods used, we characterize the perception of students by structural equations modelling (SEM). This study identifies and analyzes the factors that influence students' attitude and readiness towards mobile technology use in education, providing valuable insights into improving the adoption of new technologies and to evaluate the sustainability of m-learning in economic education.

**Keywords:** mobile-devices; m-learning; technology acceptance model; high economical education; sustainable education

## 1. Introduction

The teaching and learning pedagogy have undergone significant changes in recent decades due to technological advancements [1–3]. With technology being ubiquitous in today's society, the educational system must adapt to the needs of the new generation of students [4]. The prevalence of mobile devices and wireless connectivity has introduced a new educational paradigm, commonly referred to as mobile learning or m-learning. It comes as no surprise that teachers have been looking for ways to introduce mobile devices into traditional and electronic learning in order to enable students to learn anytime, anywhere, at their own pace, as a shift from "traditional" to "electronic" and later to "mobile" learning [5,6]. Traditional pedagogical approaches have become inefficient over time compared to students' lifestyles, so there is a need for an adaptation and evolution of

these approaches [7]. The COVID-19 pandemic has underscored the importance of flexible and intelligent educational systems. Moreover, the pandemic has also demonstrated that mobile learning has immense potential, and education systems worldwide are increasingly exploring, promoting, and embracing its new features [3,8–10]. This trend could indicate the sustainability of m-learning as an educational system.

During the pandemic, the education systems used these technological systems to ensure the continuity of the students' education [11,12]. The unexpected social isolation brought students and teachers to the situation of connecting through the mobile devices, the transition to remote teaching being realized suddenly and in an unplanned manner [13]. With regard to the education systems in the central-eastern part of Europe, which includes the present study as well, the research carried out during and after the pandemic, in different countries, highlighted the relatively common elements regarding the advantages and disadvantages of m-learning [2]. A research conducted in Romania identified several benefits such as program flexibility, adaptation to diverse learning styles, and access to numerous useful digital tools. However, concerns regarding isolation, anxiety, limited creativity, and student evaluation difficulties were also reported. Nevertheless, a significant portion of students and teachers expressed a positive attitude towards e-learning [14].

A theoretical and empirical cross-country study involving Poland, Croatia, and Serbia [10,15,16] examined the advantages and disadvantages of e-learning during and after the COVID-19 period from the perspective of students and teachers. The study concluded that while there are disadvantages, students believe that the benefits of e-learning outweigh them. Additionally, the study demonstrated that digitization and the adoption of technology-driven pedagogical approaches have been beneficial in the field of education, particularly in economic disciplines.

Preoccupations related to the factors that influenced the adoption of e-learning during COVID-19 in Hungary were also presented in the paper [17], where the analysis focused on the influence of age and gender on the desire, the intention to use e-learning tools, gender having the biggest impact. In the Czech Republic, a comparative analysis between the first and second waves of the pandemic revealed a growing positive attitude towards distance learning compared to in-person classes, aligning with the situations observed in other aforementioned countries [18].

The increasing global usage of mobile devices and the steady growth of internet penetration have led to the adoption of mobile learning. Global statistics data show that, for a population of 8.02 billion [19] at the beginning of 2023, the number of unique mobile subscribers at the end of the first quarter of 2023 was about 5.5 billion people worldwide [20]. At the same time, the number of internet users reached 5.16 billion people worldwide and 4.76 billion social media users [21]. These statistics shows that the mobile technologies are a global context and therefore, this reality cannot be ignored by the educational systems around the world, which should reflect their worldwide usage [22]. As a result, unlike conventional pedagogical approaches, education must take use of the online world because this is where young students interact and connect [23]. The paradigm shift in education is therefore mandatory to ensure future education resilience in the face of changes regarding learning and communication among students.

There is a wide range of reasons why adopting m-learning techniques is important, which includes better access to resources and learning material and the possibility of flexible teaching and learning activities, mostly to be aligned with institutional and business aims [4,24,25]. Possible changes in teaching activities points to the ways in which teachers can use mobile devices as a support element in the teaching content [2,14], while flexible learning refers to the ways in which students have the opportunity to manage their learning activities, at times and places chosen by themselves [8,15,26]. For these technologies to align with educational institutions requires that they be economically efficient by reducing the costs of expensive hardware by adopting wireless technology and mobile learning, for example, and that they provide information to a large number of students regardless of their location [27]. In addition to these arguments that relate more to the practical and technical

aspects of teaching-learning, it has been shown in various studies that m-learning has the potential, through different tools, to bring other benefits, such as strengthening cognitive motivation of students [24], of involvement and attention, etc. The benefits of m-learning were summarized in a systematic review by Saikat et al. [3] as follows: the availability of resources, the improvement of communication, the development of the students' technical skills, a better operationalization of activities and, last but not least, financial benefits. In addition, the novelty effect of different m-learning tools positively influences autonomous motivation, internalization and learning achievements of students [28]. Another argument would be that the support that mobile learning can offer to support metacognitive and cognitive processes in self-regulated learning (SRL), helping students to take control of their learning process, in order to be successful in the academic activity [26].

On the contrary, certain studies have cast doubt on the actual advantages of mobile learning and its limitations. The implementation of m-learning poses numerous challenges, some of which include limited resources, content-related issues across all subject areas (including economic education) [29,30], technical difficulties arising from inadequate knowledge and skills of instructors and students [31,32], distraction from work tasks [33], a large number of users, and the need for online connectivity [29]. The evaluation of mobile learning can also be problematic as it needs to be both transparent and secure, while ensuring that students do not receive assistance from others [34,35]. Furthermore, problems with connectivity, data protection, privacy, and confidentiality may arise in the m-learning environment, as noted by Saikat et al. [3]. One particular challenge that is relevant to our research is the inability to conduct laboratory work for courses that require it in online education. As a result, students studying medicine, engineering, and other technical or economic fields (such as accounting and commerce) are deprived of practical learning experiences in the context of mobile learning, as highlighted by Currie et al. [36].

Despite indisputable arguments regarding the usefulness of mobile technology, it is also important to asses students' readiness for mobile learning, because just owning and using mobile devices in everyday life does not necessarily mean that students are willing to use them in learning activities [37]. In light of the COVID-19 pandemic, there is a significant focus on this topic, particularly concerning the assessment of students' acceptance of m-learning and the factors that contribute to the formation of behavioral intentions. The efficacy of m-learning is currently being debated in light of the students' return to physical classrooms and face-to-face interaction with educators. If this system proves to be sustainable, questions remain about its form and the conditions under which it would be implemented. This issue has garnered considerable attention, as demonstrated by studies conducted by Al-Rahmi et al. [38], Alturki and Aldraiweesh [8], and Al-Emran et al. [39]. Such an analysis must be carried out differently on high school students, university students or for lifelong learning adults because there are essential differentiations between these categories of learners. One important aspect to consider is the availability to purchase the mobile devices, as for college students and adults this is much higher than for the high school students [40]. This must be corroborated with the learning motivation that differs markedly at various stages of academic training and after. Another argument is that universities are much more prepared and willing to introduce different applications and elements of m-learning as an official policy of institutions, than high schools, where teachers usually act individually. University autonomy allows a more efficient university management in achieving an adequate and high-performance information technology infrastructure, as well as in raising students' awareness towards new technologies [41].

The current study attempts to evaluate the students' attitude from an economical high school in Romania towards m-learning, respectively towards the use of mobile devices in learning activities. Our aim is to address the research gap identified in the field of mobile learning (m-learning) focused on economic education at the high school level. Limited studies have been conducted on this topic, with most of the existing research primarily concentrated on higher education [3,42–44]. It is crucial to note that students studying economics will play a pivotal role in the future digital economy. Therefore,

their education within this digitalized context holds significant importance. Students enrolled in this vocational (economical) high school have in general an average academical performance and average digital competences level at the national evaluations. Their core curriculum covers a multitude of economical subjects and involves many practical activities. Mobile devices are elements to consider in this context, especially since it is known that Romania has one of the best internet speed connection in the world [45] and the number of smartphone users in Romania is quite large, over 15 millions, which represents more than three quarters of the populations [46].

The research questions we would like to address are: What are the major factors that affect the attitude of economical high-school students towards the use of mobile devices in their learning activities? Are there any relationships that can be established between these factors in the economical high school education? The objective is to identify and assess these factors with the intention of developing a theoretical framework. This framework will aid in the identification of managerial strategies that can be implemented in schools to improve economic education. As a working framework to study and answer to these questions, we used the technology acceptance model which is a well-established model able to explain the determinants that influence the computer acceptance in general, as well as the impact of the external factors on attitudes and intentions to use technology in particular [47]. These questions are asked in order to know what exactly influences the students in their attitude towards the use of technology in learning in general and m-learning in particular. It is desired to identify the perspectives of their use in the future and to ensure their sustainability.

The research work is organized as follows. In Section 2 we discuss the literature review on mobile learning. In Section 3 we present the research model that we used. First, in Section 3.1 we discuss the technology acceptance model, while in Section 3.2 we outline the hypotheses to be tested in this study. In the next section we discuss the methodology and give relevant information about the participants to the study. The main results of our work are presented in Section 5. Thereafter, the conclusions, implications and limitations of this study are presented.

## 2. Literature Review on Mobile Learning

There is a wide range of research on mobile learning. With the development of the wireless technology, the field has continuously grown and is experiencing rapid evolution in the last few years, especially after the period of the COVID 19 pandemic. Upon analyzing the literature, it was observed that the terms "mobile learning" and "m-learning" are used interchangeably with a surplus of definitions. This has led to confusion and an absence of clear pedagogical theoretical framework in researches on mobile learning. Therefore, there is ongoing debate regarding the meanings of these terms and their impact on educational issues [48].

There is also an aspect related to the ambiguity of term "mobile" that has to be highlighted: The term 'mobile' refers to either mobile technologies, learner mobility or content mobility where each of these aspects has an important meaning [49]. Most conceptualizations define mobile learning from the perspective of the technological devices used and suggest that mobile learning is delivered or achieved entirely or largely through mobile technologies, even if this approach is considered too technocentric and presents some constraints [5]. Another definition given by Almaiah and Alismaiel [41] defines m-learning as a new learning technology that helps students carry out their educational activities, using mobile devices with which they can access courses, assignments, quizzes or tools evaluation. Practically, it can be observed that the term mobile learning is a topic frequently associated in research with how to use mobile devices, rather than focusing on solving educational problems, respectively improving learning performance [50].

Maybe the most important aspect that differentiate mobile learning from other pedagogical approaches is the ability for the students to perform learning activities without being tied to a certain fixed location, by using mobile devices to access and communicate

information, through wireless technology [51]. This model of distance education, using mobile devices, is very favorable and advantageous for the students, who have the opportunity to be educated independent of time and environment [52]. A study by Grant [48] synthesized the current definitions of the terms and concluded that a learning environment involving mobile learning must have certain characteristics to generate learner engagement. These include the student having mobility, being autonomous, having mobile devices available to them at all times, having data services that are always available, having content that is mobile and adapted to the students' needs and context, and incorporating tutors (embedded).

Therefore, the design and the implementation of mobile learning requires an approach in which the pedagogical and technical aspects are relevant and compatible. Thus, several studies have been conducted on the efficiency of mobile devices in the field of education, on learners' readiness, or on acceptance of mobile learning by the students [25,53,54] or by the teachers [55]. Some studies have analyzed the most relevant variables that influence the university students' attitude towards mobile technologies [56,57] or the determinants of the acceptance of mobile technologies among teachers [58]. On the technical side, other studies focus on tools and applications that can be used in mobile learning, as well as their benefits and limitations [4,24]. The use of mobile communication combined with internet tools create a stronger connection between instructor and student without increasing the pressure sometimes the student could feel from their instructors, a bond which might have the effect of increasing students' motivation [59]. Also, the inclusion in the learning process of a mobile application for student self-assessment or assessment produces an improvement in student achievement and a positive students' attitude towards new technologies [60,61]. In the literature, the mobile learning research conducted along these directions has focused on investigating the benefits and drawbacks of this type of learning.

These research directions raise technical and pedagogical cultural related issues. The sustained technological developments in the last years gradually eliminate the technological limitations of m-learning related to the small screen size of mobile devices, network speed, battery life or the limited memory of the devices. This happens even if the hardware devices and technical systems are created and marketed for corporate, retail, or recreational users and their use for educational purposes is parasitic and of secondary use [40]. Still, it is difficult to adapt a pedagogical culture to a mobile format, because this implies an adaptation of all the actors involved in the teaching and learning processes, respectively learners, instructors, curricula, educational contents and institutions [62]. Also, a bibliometric mapping shows that the most used keywords in research on mobile learning are mobile devices, mobile technologies, smartphone, tablet and higher education. Also, in recent years, the most frequently addressed topics were related to educational technologies and educational strategies [63].

Recent research on m-learning has focused on the role of mobile applications and social media in promoting critical thinking skills, comprehension, analysis, and synthesis during the learning process. However, it has been noted by Audrin and Audrin [64], Pedro et al. [32], Hosain et al. [65], and Eynon [66] that the use of these applications can also have negative effects that need to be recognized and understood. It should be noted that these applications and social media platforms are primarily designed for commercial purposes, and their use in formal education can be detrimental to learning as they are more geared towards leisure activities. Moreover, existing research on m-learning does not adequately cover the educator's perspective on the use of mobile applications, and there is a lack of theoretical and pedagogical foundations that make the integration of different m-learning strategies incompatible with the curriculum [32,67]. As a result, the orchestration of mobile devices with didactic methods and strategies requires constant adaptation, which generates a strong interest in research, irrespective of the field in which they are applied.

### 3. Research Model and the Hypotheses

*3.1. Technology Acceptance Model*

The research on incorporating new technologies in education and the study of their degree of acceptance by students and teachers led to the issuance and use of theories and models with a great potential for analyzing the different types of technologies adopted in learning. Thus, some of the most important models are: Technology Acceptance Model (TAM), Decomposed Theory of Planned Behavior (DTPB) or Unified Theory of Acceptance and Use of Technology (UTAUT) [1]. In this work, we adopted TAM in order to evaluate the perception of economic high school students on the use of m-learning. Technology acceptance model (TAM) was introduced by Davis [68] and has enjoyed great popularity since its publication. This model is an adaptation of theory of reasoned action (TRA) and it was introduced as a framework to explain how users come to accept and use technology. The model aims to analyze the determinants of computer acceptance as well as their influence on user behavior of computing technologies. Furthermore, TAM assesses the influence of external factors on internal beliefs, attitudes and intention to use. Among the primary beliefs considered to have the greatest impact on the attitude and the behavior of technology users, the most important ones highlighted by the TAM model are perceived usefulness and perceived ease of use [47,69]. Although TAM uses TRA as the basis for establishing the relations between model elements, it is less general than TRA, being designed only to analyze the behavior of computer users [55]. TAM shows that, in order to make people able to use technology, we need to produce behavioral intention. To produce this intention of behavior in relation to technology, users must adopt a certain attitude towards the use of technology. Besides, the perceived usefulness and perceived ease of use, the attitude is influenced by a number of other external factors that influence attitudes towards technology. Since its original proposal, the TAM model has been used on various research topics. In this context, improved variants of the model have appeared. Thus, the two major variants are TAM 2 [70] and TAM 3, a model that was developed initially for e-commerce [71].

Currently TAM is used to explain the process of adopting technology in various fields of activity in which technology is essential. In particular, in e-commerce, TAM was adapted and operationalized for instant shopping, showing that perceived enjoyment has a particularly strong impact on the intention to use [72] or for understanding the consumer's acceptance of e-shopping, considering e-shopping quality, enjoyment and trust as TAM components [73]. In the healthcare field, the implementation of technology is essential for the development of new therapies or for system management, as well as to evaluate the acceptance of technology for the medical system users. In that regard, the use of TAM to assess the acceptance of information technology in the context of health information management founds that users' perception of usefulness and ease of use remain the most important determinants of acceptance of information technologies [74].

Due to the diverse applicability of information and communications technology (ICT) in the educational field, TAM is also used to analyze technology acceptance of both students and teachers, considering technology acceptance as a key element of the process of integrating ICT in the educational process. There are many studies that use TAM starting with primary education, high-school education, but the most investigated level is the university one. For example, using TAM along with the theory of planned behavior (TPB) offers a comprehensive understanding of the factors that significantly affect the college students' attitude toward a cloud computing classroom [75]. Another example of research, based on TAM and using UTAUT, analyzes whether there are gender or age differences when it comes to the acceptance of m-learning by the students [76]. It was found that age differences moderate the effects of effort expectancy and social influence on m-learning use intention. Researchers have given significant attention to the analysis of smartphones' role in mobile learning, with Hartley and Andújar's [4] work indicating that smartphones will have a crucial role in learning activities, necessitating systematic training based on this device. The use of social media platforms [77] or virtual reality [78] in education has also been evaluated

to determine their impact as innovative technologies, highlighting their positive aspects as well as the challenges that arise from their implementation. The results of another research suggest that two of the main constructs of TAM, perceived usefulness and attitude, have a major influence on students' acceptance of m-learning [53]. As a final example, TAM was used to show that the acceptance of mobile technology as learning tool by the students have a great impact on their learning achievement and motivation [41,57,79,80].

Many studies focused on the acceptance of the new technologies among teachers as well. Thus, applying TAM to perform a path analysis of the determinants of pre-service teachers attitudes to computers, shows that some external variables used to extend the original TAM, respectively subjective norm, facilitating conditions and technological complexity are significant determinants [55]. In another study, the effect of some external variables, namely, previous experience, perceived enjoyment, self-efficacy, facilitating conditions and subjective norm has been asses within the TAM model to investigate the teachers' intention of using mobile technologies [58].

### 3.2. Aim and Hypotheses

This study aims to apply TAM to a group of students from an economical high school to determine and evaluate the determinants of their attitude towards the use of mobile devices in an area of research that has not been widely addressed so far. Researches that used TAM in the last years had introduced a number of external variables into the model to investigate attitudes toward computer use. In this paper we used as external variables the subjective norm [81] and facilitating conditions [82] as environmental factors and learning autonomy [62] and self-efficacy [83] as technical factors. The two categories of factors are decisive for the students' attitude towards the use of mobile technology and for its embrace in the learning activity. In Figure 1 we present the list of constructors used in this study and indicates the list of the hypotheses proposed to be tested. Below we describe each constructor independently and discuss the associated hypotheses.

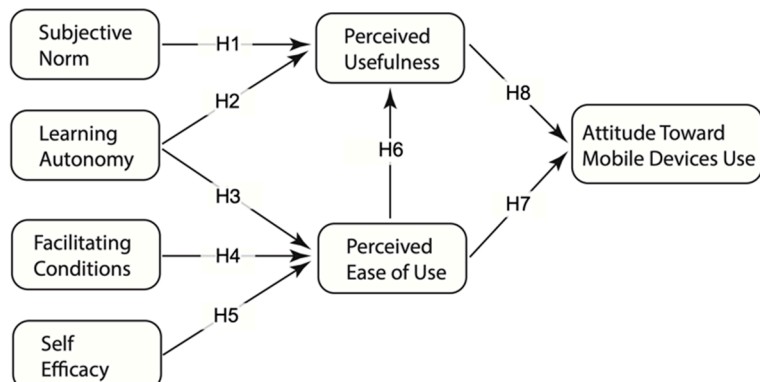

**Figure 1.** Graphical representation of the research model used in this study, showing the list of constructors and the corresponding working hypotheses.

*Subjective Norm.* The subjective norm (SN) constructor was first introduced in theory of reasoned action being defined as "the person's perception that most people who are important to him think he should or should not perform the behavior in question" [81]. This variable, frequently introduced in various researches that use TAM, refers to the pressure that society puts on the individual to adopt a certain behavior. In this work it is about the social and organizational pressure laid on students to use mobile devices for the purpose of learning, not just for entertainment. In a similar study, subjective norm was used in a model that assess the acceptance of mobile technologies among teachers, having a positive influence on perceived usefulness and behavioral intention [58]. The same constructor was used, for example, to evaluate students' acceptance of a learning platform in the study of statistics in a blended learning context, using simulations, online videos and online quizzes as digital resources [84].

**Hypothesis (H1).** *High school students' subjective norm positively influences the perceived usefulness that students feel when using mobile devices as learning tools.*

*Learning Autonomy*. The learning autonomy (LA) construct refers to the situation in which students have control over their learning process that use mobile devices and are responsible for how this process is performed. Learning autonomy has been demonstrated to have a significant positive influence on students' behavioral control for m-learning in a study conduct for investigate the mobile learning readiness in higher education [62]. Autonomy has a major impact on student satisfaction because it has been shown that motivation based on autonomy has a stronger impact on satisfaction than external motivation [85]. Another study on students' behavioral intention regarding an Open Source Software (OSS) shows that the external variable autonomy has a positive effect on perceived usefulness and perceived ease of use [86].

**Hypothesis (H2).** *High school students' learning autonomy positively influences the perceived usefulness of using mobile devices.*

**Hypothesis (H3).** *High school students' learning autonomy positively influences the perceived ease of using the mobile devices by the students.*

*Facilitating Conditions*. In order to better understand the factors that influence the use of the computer, Triandis [82] has developed a new model, called the theory of behavior, by modifying the theory of reasoned action. In his model a new construct, called facilitating conditions (FC) was introduced defined as the "objective factors, 'out there' in the environment, that several judges or observers can agree to make an act easy to do". Regarding the use of mobile devices by students, facilitating conditions would present the provision of support for students for using technology, by training and assisting them when they encounter difficulties. Regarding m-learning, it has been shown that facilitating conditions, i.e., the resources provided, the technical support, the available trainings have a strong influence on the involvement of students in m-learning programs [87].

**Hypothesis (H4).** *High school students' facilitating conditions positively influences the perceived ease of use.*

*Self-efficacy*. This term was introduced by Bandura [83] and describes the confidence that a person has in his ability to perform certain tasks and skills, and the belief that can achieve what sets out to do. Self-efficacy (SE) beliefs induce the way people feel, think, motivate themselves and behave. Thus, people who pay more attention to being competent in a certain work, able to perform certain tasks, are likely to have a greater intention to execute that work [88]. The analysis of self-efficacy has been approached in several researches. For example, in a study that had students in an accounting course as subjects, the analysis showed that increasing self-efficacy in students leads to improved academic achievement [89]. Moreover, it was shown that there is a direct link between students' self-efficacy in the use of various technological devices and success in learning [90]. Also, another study that used factor analysis showed that self-efficacy is an antecedent to students' online learning acceptance [91].

**Hypothesis (H5).** *High school students' self-efficacy positively influences the perceived ease of use that students feel when using mobile devices as learning tools.*

*Perceived Ease of Use.* This constructor was introduced in the original TAM model by Davis et al. [47] together with perceived usefulness as key determinants of the attitude towards the use of technology. In the original model, the perceived ease of use (PEU) refers to "the degree to which the prospective user expects the target system to be free of effort" [47]. Also, in addition to the positive effect it has on attitude, the perceived ease

of use positively influences perceived usefulness. These connections between constructs have been addressed in numerous studies, some related to the intention to use m-learning. Thus, in the work of Al-Emran et al. [39] showed that PEU is an important predictor that explains the continuous intention to use mobile learning, which in turn is influenced by the students' attitude towards this type of learning. At the same time, the same construct positively influences perceived usefulness. The same correlations were also shown in two other studies related to the factors that influence students' acceptance of m-learning in university study programs [54,80].

**Hypothesis (H6).** *High school students' perceived ease of use positively influences the perceived usefulness of using mobile devices.*

**Hypothesis (H7).** *High school students' perceived ease of use positively influences their attitude toward mobile devices use.*

*Perceived Usefulness.* Perceived usefulness (PU) is defined as "prospective user's subjective probability that using a specific application system will increase his or her job performance within an organizational context" [47]. This increase in performance refers to the speed and the efficiency in accomplishing work tasks and these aspects are taking into account when the user decides to use a certain mobile device. For example, Teo [56] analyses the effect of perceived usefulness on pre-service teacher's attitudes to computer use and showed that this is a key determinant of attitude, directly and significantly influencing teachers' attitudes toward computer use. It was also shown, in a study regarding the intention to use m-learning by a group of postgraduates of The British University in Dubai in the United Arab of Emirates (UAE) that there is a positive strong correlation between perceived usefulness and satisfaction. This connection was also confirmed by other research [92,93] and is due to the fact that when students see the usefulness of using m-learning by increasing their performance, they will experience high levels of satisfaction, which will affect their attitude towards the use of m-learning [39]. Our hypothesis related to the role of PU is:

**Hypothesis (H8).** *High school students' perceived usefulness significantly and positively influences their attitude toward mobile devices use.*

*Attitude Toward Mobile Devices Use.* In literature related to TAM, attitude is defined as the emotional reaction that occurs when individuals use technology and that can manifest itself in the form of positive or negative feelings related to the performance of certain tasks [57]. In another definition, attitude is described as "one's desirability to use the system" [94]. In order to adopt a certain behavior towards technology, individuals must form a certain attitude towards the use of technology. Attitude is therefore a significant factor in determining how information technology is used [81]. Numerous researches have shown that the attitude towards m-learning has a major contribution to the development of a behavior that includes m-learning systems as desirable tools [39,95].

## 4. Methodology, Data Collection and Data Analysis

The participants in this study were 407 students from an economical public high school situated in Oradea, Romania, an important school, with over 1100 students. The profile of the qualifications from this school is 'services' in the field of economy, administration, trade and tourism. The core curricula include disciplines covering various fields of qualification such as applied economics, accounting, finance, company management, human resources, marketing, insurance, statistics, hotel management and many others. Also, the students go through the ICT discipline every year during the four years of high school period, the digital competences being evaluated at the high school final exams. The sample was thought to be representative of the entire school population included in this study. Thus, boys and girls students, from all study levels, as seen in Table 1 and from all the 4 profiles

mentioned above, were considered. All participants have at least one mobile device, and some of them use more than one device in their learning activity. As displayed in Table 1 The most common devices owned and used are smartphones (388 students) and laptops (237 students).

**Table 1.** Information about participants (407 students).

| | Grade | | | | Gender | | Mobile Devices Used by the Students | | | |
|---|---|---|---|---|---|---|---|---|---|---|
| IX | X | XI | XII | Girls | Boys | Smart-Phone | Tablet | Laptop | Others |
| 119 29.24 (%) | 92 22.60 (%) | 88 21.62 (%) | 108 26.54 (%) | 259 63.64 (%) | 148 36.36 (%) | 388 | 171 | 237 | 16 |

The data was collected by a multiple item online survey in the period April-May 2022. In this sense, a Google Form was used, the link of which was distributed to the students through the school teachers, specifying the non-obligatory, voluntary character of the participation in the research. Considering the number of classes in which participation was requested, more than 500 students were considered, but in the end, there were only 407 valid answers. The questionnaire contains 23 items related to 7 constructs, designed and adapted according to other studies previously conducted by various researchers (see Appendix A). For attitudes towards mobile devices use, learning autonomy, subjective norm, facilitating conditions, self-efficacy, 3 items each were designed, and for perceived usefulness, perceived ease of use 4 items each. To test the quality of the questionnaire and its comprehensibility, a pre-test was first applied to a class of students. The ambiguities in the identified statements were corrected, reformulations were made to increase the level of understanding, and the Google Forms used were also subjected to a critical evaluation and form changes. To assess students' perceptions of the use of technology, a 5-point Likert scale was used, ranging from totally disagree to totally agree. This survey includes three parts, the first briefly explains the purpose of this study, the second requires general information (gender, class, mobile devices used in learning activities). In the third part, the students actually completed the survey regarding their attitude towards the use of mobile devices in their learning activity. On average, a student needed 10 min to complete the questionnaire.

In this study, the maximum likelihood estimator was employed to test the hypotheses and evaluate the model. The structural equation modeling (SEM) approach, implemented using the Lavaan package [96], was utilized for this purpose. Reliability analysis, including measures such as Cronbach's alpha, was conducted on the collected data to assess the consistency of the constructs. Additionally, other comparative indices such as the Comparative Fit Index (CFI) or the Tucker-Lewis Index (TLI) were considered during the analysis. The results indicated that all values exceeded 0.70, which is considered acceptable [97–100]. Discriminant validity was also examined, revealing that the average variance extracted (AVE) values exceeded the correlation values between variables. Based on the SEM analysis, all eight hypotheses in the research model were confirmed. The details of our findings are presented in the next section.

## 5. Results and Discussion

The main purpose of this study is to investigate if the extension of the TAM model discussed in Section 3.2 is suitable to illustrate the attitude toward mobile learning of the economical high school students. In addition to the regular constructors used in TAM we have consider other supplemental external factors and analyze their implications.

The quality of the solution is captured through a list of fitting indices that established whether the model is acceptable or not. The most important indices are displayed in Table 2 together with their recommended values. The most common ones such as the comparative fit index (CFI = 0.933) and the Tucker-Lewis index (TLI = 0.922) as well as the root mean square error of approximation (RMSEA = 0.064) are in favor of the theoretical proposed

model. Furthermore, the other indices displayed in Table 2 are within recommended intervals which confirm the validity of our model.

**Table 2.** Information about participants (407 students). Model fit indices for the TAM model graphically displayed in Figure 1.

| Fitted Index | Value | Level of Acceptance | References |
|:---:|:---:|:---:|:---:|
| $\chi^2$ | 631.848 | irrelevant | [101] |
| $\chi^2/\mathrm{df}$ | 2.92 | $\leq 3$ | [97,101] |
| CFI | 0.933 | $\geq 0.9$ | [98,99] |
| TLI | 0.922 | $\geq 0.9$ | [97,100] |
| IFI | 0.927 | $\geq 0.9$ | [101,102] |
| RMSEA | 0.064 | $<0.8$ | [98,103] |
| RMR | 0.058 | $<0.8$ | [104] |
| SRMR | 0.074 | $<0.8$ | [105] |

To validate the reliability of the constructors, we present the Cronbach's $\alpha$-coefficients in Table 3. They express the consistency of our constructors [106]. As proposed by Gefen, Straub and Boudreau [107] all the values are larger than 0.70, indicating satisfying reliability. In the same table we also display the composite reliability coefficients $\omega$'s as proposed by Raykov [108] which can be considered as an alternative estimates of Cronbach's $\alpha$'s with a similar acceptability threshold of 0.70. Table 3 also presents the standard factors loading values. They show the importance of the relationship between the constructs and the observed variables in our survey and in our case, all exceeds 0.5. A separate column displays the average variance extracted (AVE) for each construct in the model [109]. Values above 0.5 are considered in general to be acceptable, while levels above 0.7 are very good. According to Fornell and Larcker [109] the assessment of discriminant validity, which consists in comparing the amount of variance captured by a construct $\sqrt{AVE}$, with the shared correlations with other constructs has become maybe the most widely used method for this purpose. Such a comparison is presented in Table 4, which indicates that $\sqrt{AVE}$ is always greater than the correlation involving that particular construct.

In the TAM version that we consider, a set of 4 external factors were proposed, consisting of Subjective norm, Learning Autonomy, Facilitating conditions and Self-Efficacy. Figure 2 and Table 5 showcase the primary outcome of our analysis, revealing the path coefficients of the model together with the corresponding *t*-values and *p*-values [110]. These coefficients substantiate the validity of the proposed hypotheses and the associations between the constructs. Our findings provide validation for the underlying TAM model, indicating that perceived usefulness ($\beta \approx 0.642$, $t = 9.353$, $p < 0.001$) has the most significant impact on ATMDU, followed by perceived ease of use ($\beta \approx 0.190$, $t = 2.745$, $p = 0.006$).

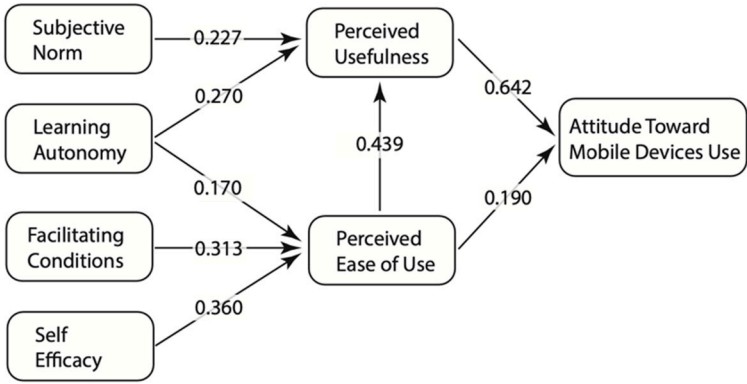

**Figure 2.** Path analysis for the research model introduced in Figure 1.

**Table 3.** Model fit indices.

| Fitted Index | Mean | Standard Deviation | Factor Loadings | Cronbach's α | AVE | ω | $R^2$ |
|---|---|---|---|---|---|---|---|
| Attitudes Towards Mobile Devices Use | | | | 0.874 | 0.705 | 0.877 | 0.613 |
| ATMDU1 | 3.82 | 1.01 | 0.876 | | | | |
| ATMDU2 | 3.77 | 1.00 | 0.883 | | | | |
| ATMDU3 | 3.79 | 1.03 | 0.760 | | | | |
| Perceived Usefulness | | | | 0.871 | 0.624 | 0.868 | 0.574 |
| PU1 | 3.79 | 0.91 | 0.826 | | | | |
| PU2 | 3.98 | 0.93 | 0.700 | | | | |
| PU3 | 3.54 | 0.95 | 0.809 | | | | |
| PU4 | 3.90 | 0.82 | 0.828 | | | | |
| Perceived Ease of Use | | | | 0.816 | 0.550 | 0.826 | 0.579 |
| PEU1 | 4.05 | 0.88 | 0.796 | | | | |
| PEU2 | 3.94 | 0.85 | 0.802 | | | | |
| PEU3 | 4.21 | 0.80 | 0.717 | | | | |
| PEU4 | 4.28 | 0.76 | 0.599 | | | | |
| Learning Autonomy | | | | 0.838 | 0.681 | 0.861 | |
| LA1 | 4.19 | 0.83 | 0.643 | | | | 0.413 |
| LA2 | 3.90 | 0.94 | 0.877 | | | | 0.769 |
| LA3 | 4.23 | 0.76 | 0.930 | | | | 0.864 |
| Subjective Norm | | | | 0.740 | 0.524 | 0.762 | |
| SN1 | 3.33 | 0.89 | 0.844 | | | | 0.713 |
| SN2 | 3.35 | 0.98 | 0.513 | | | | 0.263 |
| SN3 | 3.23 | 0.90 | 0.805 | | | | 0.647 |
| Facilitating Conditions | | | | 0.809 | 0.588 | 0.810 | |
| FC1 | 4.26 | 0.73 | 0.779 | | | | 0.607 |
| FC2 | 4.27 | 0.80 | 0.778 | | | | 0.605 |
| FC3 | 3.96 | 0.85 | 0.748 | | | | 0.559 |
| Self-Efficacy | | | | 0.862 | 0.677 | 0.862 | |
| SE1 | 4.34 | 0.76 | 0.779 | | | | 0.607 |
| SE2 | 4.25 | 0.76 | 0.839 | | | | 0.703 |
| SE3 | 4.26 | 0.78 | 0.849 | | | | 0.720 |

**Table 4.** Discriminant validity for the measurement model [109]. The diagonal elements represent the $\sqrt{AVE}$ of the values listed in Table 3, while the off-diagonal elements are the correlation between various constructs. The diagonal entries are larger than the off-diagonal elements.

| Construct | ATMDU | PU | PEU | LA | SN | FC | SE |
|---|---|---|---|---|---|---|---|
| ATMDU | 0.840 | | | | | | |
| PU | 0.770 | 0.790 | | | | | |
| PEU | 0.624 | 0.675 | 0.742 | | | | |
| LA | 0.519 | 0.628 | 0.609 | 0.825 | | | |
| SN | 0.363 | 0.473 | 0.314 | 0.398 | 0.724 | | |
| FC | 0.507 | 0.582 | 0.702 | 0.663 | 0.416 | 0.767 | |
| SE | 0.493 | 0.558 | 0.709 | 0.641 | 0.321 | 0.767 | 0.823 |

Thus, hypotheses H7, H8 are confirmed. These results are in line with the previously re-realized researches regarding core TAM variables (PEU and PU). The importance of these two constructors in terms of m-learning has been investigated by numerous researchers [38,57,80,90], and the results of this study are compatible with their results. The impact of these two factors (PEU and PU) on the use of mobile devices can therefore be observed, and their increase leads to a better use of m-learning [8]. Students also use m-learning because they are familiar with mobile technical devices. If they perceive

an ease in their use, this leads to an improvement in the attitude and intention to use m-learning to enhance learning and knowledge.

**Table 5.** Hypotheses testing results.

| Hypotheses | Relationship | β-Coefficient | *t*-Value | *p*-Value | Remarks |
|---|---|---|---|---|---|
| H1 | SN → PU | 0.227 | 4.394 | 0.000 | Supported |
| H2 | LA → PU | 0.270 | 3.488 | 0.000 | Supported |
| H3 | LA → PEU | 0.170 | 2.167 | 0.030 | Supported |
| H4 | FC → PEU | 0.313 | 3.317 | 0.000 | Supported |
| H5 | SE → PEU | 0.360 | 3.823 | 0.000 | Supported |
| H6 | PEU → PU | 0.439 | 6.098 | 0.000 | Supported |
| H7 | PEU → ATMDU | 0.190 | 2.745 | 0.006 | Supported |
| H8 | PU → ATMDU | 0.642 | 9.353 | 0.000 | Supported |

Another result of the research indicates that perceived ease of use has a strong positive influence on perceived usefulness ($\beta \approx 0.439$, $t = 6.098$, $p < 0.001$), as we saw presented in the previously mentioned TAM literature, thus confirming hypothesis H6. These results align with what was shown by Senaratne et al. [111], Joo et al. [93], Sabah [112] and shows that when students perceive an ease in using mobile devices in m-learning they will feel as a consequence a greater sense of the benefit of m-learning [80].

The results support also the other hypotheses (H1–H5). It appears that the most relevant contribution to perceived ease of use is due to self-efficacy ($\beta \approx 0.360$, t = 3.823, $p < 0.001$), followed by facilitating conditions ($\beta \approx 0.313$, $t = 3.317$, $p = 0.001$) and learning autonomy ($\beta \approx 0.170$, $t = 2.167$, $p = 0.03$). On the other hand, subjective norm seems to have a slightly smaller effect ($\beta \approx 0.227$, $t = 4.394$, $p < 0.001$) on the perceived usefulness as compared to learning autonomy ($\beta \approx 0.270$, $t = 3.488$, $p < 0.001$). We have analyzed the impact of these four external factors because they are among the most important predictors of the two core Technology Acceptance Model (TAM) variables (Perceived Usefulness—PU and Perceived Ease of Use—PEU) that have been addressed in previous research and are considered to have the greatest impact [1]. Our study reveals that self-efficacy has the strongest impact on perceived ease of use. This finding is consistent with what other research has shown regarding the significant impact of perceived ease of use [111,113,114]. This can be interpreted in light of the fact that students who have confidence in their mobile device usage skills and expertise will experience a sense of ease in using m-learning and will be more inclined to adopt it. The next influential factor on perceived ease of use in this research is facilitating conditions. This finding aligns with what other studies [87,115,116] have shown regarding the importance of technical infrastructure provided by educational institutions and the support teachers should offer students during m-learning activities. This factor can be further analyzed by breaking it down into different components, as demonstrated by Almaiah et al. [9], who addressed variables such as IT infrastructure and university management support.

These results indicate that students who have access to technical support, receive assistance through technical resources and training, and are supported when facing difficulties, will experience greater ease in using m-learning. The third variable that influences perceived ease of use is learning autonomy. It also has a positive impact on perceived usefulness, but with a stronger influence. This result can be explained by the fact that autonomy in learning is highly important in distance learning environments, and it is encouraged [117]. The obtained results demonstrate that taking control over one's own learning process increases students' motivation and subsequently enhances the level of ease in using and perceiving the usefulness of mobile devices [93]. Another explanation for the results is that autonomous students are eager to actively participate in the learning process in order to acquire knowledge gradually [118]. Regarding the variable of subjective norm included in the research, it directly influences perceived usefulness. Similar contributions have been shown in other studies [119,120]. The results can be interpreted

as students understanding that important individuals in their lives, such as parents and teachers, expect them to utilize mobile devices for learning purposes, indicating the utility of these tools beyond mere entertainment.

## 6. Conclusions

In the present work we extend the technology acceptance model to analyze the determinants that influence the attitude of high school students from an economical high school towards the use of mobile devices in m-learning. We explored how the attitude toward mobile devices use is influenced by the perceived usefulness and perceived ease of use, as part of the TAM, and investigate the role played by other external variables and their effects on the attitude. In particular, we study the role of learning autonomy, subjective norm, facilitating conditions and the self-efficacy.

The results of our study reveal the following findings: (i) Validation of the TAM model: We observed positive relationships between perceived usefulness, perceived ease of use, and attitude toward mobile device use among high school students. This confirms our initial hypotheses that perceived ease of use positively influences the perceived usefulness of using mobile devices, as well as students' attitude toward mobile device use. Additionally, we found that perceived usefulness has a significant and positive impact on students' attitude toward mobile device use. Furthermore, the path coefficients indicate that perceived usefulness has a greater influence than perceived ease of use on attitude toward mobile device use. (ii) Positive relationships between perceived usefulness and subjective norm, as well as learning autonomy, with similar impacts: Our findings demonstrate that perceived usefulness is positively related to both subjective norm and learning autonomy, with comparable effects. (iii) Positive relationships between perceived ease of use and learning autonomy, facilitating conditions, and self-efficacy: Our results indicate that perceived ease of use has positive associations with learning autonomy, facilitating conditions, and self-efficacy. Notably, self-efficacy has the most significant impact on perceived ease of use, followed by facilitating conditions, confirming the hypotheses formulated in our research.

Therefore, this study validates the TAM concept regarding the use of mobile devices and reveals the impact of the proposed external factors. The obtained results in this paper fill a gap in the existing literature. In the field of economic education, there is interest in m-learning at the university level [3,42,44,121,122], but very little attention has been given to secondary education, particularly in the post-pandemic period after returning to classrooms. Previous cross-country study (Poland, Croatia, Serbia) [2], addresses the topic of economic education through e-learning at the high school level, and this present research can contribute to the development of knowledge from a broader perspective, expanding the scope of observation to obtain a more comprehensive understanding of students' attitudes towards m-learning.

The contributions of this study to m-learning acceptance research is valuable in that it applies to students studying in an economical high school, a target group that has been less analyzed from this perspective. There is a rich literature that evaluates the degree of acceptance of mobile learning in higher education, but in upper secondary school there is a major attention gap on this topic. As we mentioned before, the attitude and motivation of students towards m-learning are different in high school or at the university level. These depend both on individual financial aspects, the motivation for higher education among university students or on the institutionally available technical infrastructure, obviously much better financed in higher education.

Furthermore, this study is specifically aimed at a specific group of students who are pursuing economic subjects. The field of economics inherently offers various opportunities for utilizing mobile learning, such as simulation platforms and economic strategies, e-quizzes, online courses, role-playing games/avatars, online competitions, and gamified platforms. However, it is worth noting that the attention given to this particular category of students is currently limited. Therefore, the objective of this research is to contribute to our understanding of how students in this group either embrace or reject new technologies in

mobile learning. This understanding is crucial because the current circumstances highlight the necessity for strategic implementation plans for mobile learning, which should be developed by public authorities responsible for education, educational institutions, and other relevant stakeholders. The ultimate goal of these plans is to enhance students' readiness and acceptance of mobile learning. It is expected of today's students to have confidence in their ability to use technology for knowledge and skill development, as this is a societal expectation of their proficiency upon completion of their studies and entry into the workforce. This process of implementing m-learning is accelerated by the growing importance of global information communication technologies and the need for a viable alternative to face-to-face education in various situations, such as pandemic periods. At the same time, an adaptation of the pedagogical culture to a mobile format must be considered. These factors collectively suggest that m-learning is sustainable and the utilization of different technological devices are here to stay, paving the way for novel forms, tools, approaches, and instructional designs that will shape the future of education.

Regarding practical implications, this study contributes to the research on how economic education should be delivered at the high school level. The conclusions indicate that schools need to adapt their courses, both in terms of content and tools, to the m-learning format in order to encourage students to use mobile devices for learning purposes, rather than just for entertainment. Additionally, schools should foster learning autonomy among students, empowering them to take control of their own learning process, as this has a significant impact on their satisfaction and perceived usefulness of mobile devices. At the same time, schools need to provide favorable conditions for students to use mobile devices, including technical support and specialized instruction, so that they perceive ease in their usage and develop a positive attitude towards them. The development of ICT competencies in students and fostering their confidence in performing tasks involving these competencies are additional recommendations based on the findings of this study. Educational institutions should cultivate a sense of utility regarding the use of m-learning through various measures, which will lead to improved performance and increased satisfaction in learning. Furthermore, this study can serve as a foundation for future research on teachers' attitudes towards the use of mobile devices and m-learning.

One limitation of this study is given by the existence of other external variables that influence the attitude toward mobile devices use, which limits the generalization of the model. We are planning to address this issue in our future research. Additionally, another limitation is the size of the target group in our research; an extended analysis with a larger group that includes nationally representative economic colleges will be of interest in the future. We believe that the findings of this study should help to design effective m-learning systems for high school students in economical fields and not only.

**Author Contributions:** Conceptualization, M.M. and A.B.; methodology, M.M.; software, M.M.; validation, M.M.; investigation, M.M.; resources, M.M.; writing—original draft preparation, M.M.; writing—review and editing, A.B. All authors have read and agreed to the published version of the manuscript.

**Funding:** This work has been partially supported by the University of Oradea, within the Grants Competition "Scientific Research of Excellence Related to Priority Areas with Capitalization through Technology Transfer: INO—TRANSFER—UO—2nd Edition ", Project No. 233/2022.

**Institutional Review Board Statement:** Not applicable.

**Informed Consent Statement:** Informed consent was obtained from all subjects involved in the study.

**Data Availability Statement:** Not Applicable.

**Conflicts of Interest:** The authors declare no conflict of interest.

**Appendix A**

In this appendix we list the questions that were addressed to the high school students in this survey. Some of the questions were adapted from previous studies where the respondents were in general teachers, instructors or university students.

*Appendix A.1. Attitudes towards Mobile Devices Use [57,121]*

ATMDU1 Mobile devices make learning more interesting.
ATMDU2 Learning using mobile devices is fun.
ATMDU3 I look forward to those learning activities that require me to use mobile devices.

*Appendix A.2. Perceived Usefulness [39,47,62]*

PU1 I believe that using mobile devices would improve my performance in learning process.
PU2 I believe that using mobile devices in my learning activity would allow me to get my work done more quickly.
PU3 Using mobile devices would enhance my learning effectiveness.
PU4 I find mobile devices as useful tools in my learning activities.

*Appendix A.3. Perceived Ease of Use [47,58,62]*

PEU1 Learning to operate mobile devices would be easy for me.
PEU2 It would be easy for me to get good mobile devices usage skills.
PEU3 I believe it would be easy to access educational materials with my mobile devices.
PEU4 I find it easy to interact with mobile devices.

*Appendix A.4. Learning Autonomy [62,86]*

LA1 I would be able to access educational materials using mobile devices at any time.
LA2 I would be able to control the pace of my learning activities using mobile devices.
LA3 I would have more opportunities to perform my learning task using mobile devices.

*Appendix A.5. Subjective Norm [58,84]*

SN1 People whose opinions I value (Instructors, teachers) will encourage me to use mobile devices.
SN2 My parents expect me to use mobile devices as a learning aid.
SN3 People whose opinions I value (Instructors, teachers) will support me to use mobile devices.

*Appendix A.6. Facilitating Conditions [58,87]*

FC1 When I need help to use mobile devices specialised instructions are available to me.
FC2 I have the necessary technical resources to participate in educational activities delivered through mobile devices.
FC3 Specialized instruction regarding the use of popular educational software (apps) is available to me.

*Appendix A.7. Self-Efficacy [58,88]*

SE1 I know how to use mobile devices even if no one has taught me.
SE2 I have enough skills on how to use efficiently the mobile devices in my learning activities.
SE3 I can use mobile devices in my activities even if there's no one to help me.

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
