# Peer review of "Determinants of Economical High School Students’ Attitudes toward Mobile Devices Use"

_sustainability, doi:10.3390/su15129331_

Round 1

Reviewer 1 Report

The manuscript entitled "The Perception of Economical High School Students Toward M-Learning"  is well-designed and well-executed. The authors provide a clear research question, a detailed methodology, and a thorough analysis of the data collected. The findings are presented in a logical and organized manner. 

The following suggestion needs to be incorporated into the manuscript:

Please, add more literature on e-learning from your region during the Covid-19, such as:

Sustainability Analysis of the E-Learning Education System during Pandemic Period—COVID-19 in Romania (https://doi.org/10.3390/su12219030)

Student Performance on an E-Learning Platform: Mixed Method Approach (https://doi.org/10.3991/ijet.v15i02.11646)

Modelling Students’ Adoption of E-Learning During the

COVID-19 Pandemic: Hungarian Perspective (https://doi.org/10.3991/ijet.v17i07.29243)

The main suggestion to authors is to better discuss the main finding of the study:

Theoretical implications to explain how your research fills the gap in the previous literature.

Practical implications to explain how your research could help education institutions to better organize their lectures.

Overall, "The Perception of Economical High School Students Toward M-Learning" is a valuable contribution to the literature. While there are some limitations to the study, they do not detract significantly from its overall quality. Therefore, I recommend a revision, which should include expanding the discussion parts and consulting some previous literature.

Minor editing of English language required.

Author Response

Attached File: 

Response to Referee 1

We would like to thank the Referee for the positive report and constructive comments. We are especially glad that she/he finds our work “valuable contribution to the literature”. Below, we provide our responses to the report. Texts quoted from the referee reports are in black, while our responses are in blue. In the revised manuscript, all the modifications are indicated in red for easy identification.

Please, add more literature on e-learning from your region during the Covid-19, such as:

Sustainability Analysis of the E-Learning Education System during Pandemic Period—COVID-19 in Romania (https://doi.org/10.3390/su12219030)

Student Performance on an E-Learning Platform: Mixed Method Approach (https://doi.org/10.3991/ijet.v15i02.11646)

Modelling Students’ Adoption of E-Learning During the COVID-19 Pandemic: Hungarian Perspective (https://doi.org/10.3991/ijet.v17i07.29243)

We thank the Referee for pointing out this issue. Including information about the m-learning situation during the Covid-19 period in the region is useful as it helps to present a broader picture of how education systems incorporate m-learning. During the pandemic, students and teachers were suddenly faced with challenges related to the intensive use of mobile devices and technologies as a substitute for face-to-face schooling. Analyzing their reactions in this context is valuable for understanding the long-term impact of e-learning. In this regard, the references suggested by the Referee are extremely helpful, and we have incorporated them into the paper along with other relevant works identified on the same topic. The paragraph discussing this aspect can be found in the revised manuscript along the lines 52-78. Following the Referee suggestion we have included about ten references in the revised version.

The main suggestion to authors is to better discuss the main finding of the study:

We agree that the initial version of the paper did not provide a clear explanation of the main findings, primarily due to the manuscript's length, as we aimed to focus on presenting the main results. Consequently, in the revised version, we have significantly improved the Results and Discussion section by expanding the discussion of the results. This was done to ensure a comprehensive understanding of the study.

Theoretical implications to explain how your research fills the gap in the previous literature.

The results obtained in this study fill a gap in the existing literature. In the field of economic education at the university level, there is a high interest and abundant literature on m-learning (Ref. [3,42,44,122,123] in the revised manuscript). However, very little attention has been given to secondary education, especially in the post-pandemic period, after returning to the classrooms. The cross-country study (across Poland, Croatia, Serbia, Ref. [10,15,16] addresses the topic of economic education in the e-learning format at the high school level. Therefore, this present research can contribute to the broader understanding of students' attitudes towards m-learning and the factors influencing these attitudes. By expanding the scope of observation, it provides a more comprehensive picture of the topic. For that we have included a discussion in the Results and Discussions section and in Conclusions (lines 622-631, 641-652).

Practical implications to explain how your research could help education institutions to better organize their lectures.

Thank you again for pointing us towards this important observation. Regarding the practical implications, this study contributes to research on how economic education should be delivered at the high school level. Our findings indicate that schools need to adapt their courses, both in terms of content and tools, to the m-learning format. This will enable students to use mobile devices for learning purposes, rather than just for entertainment, thereby impacting their perceived usefulness and attitudes towards mobile devices. Additionally, schools should foster learning autonomy among students, allowing them to have control over their own learning process, as this has a significant impact on their satisfaction and perceived usefulness when using mobile devices. Furthermore, schools should provide conducive conditions for students to use mobile devices, ensuring ease of use and fostering a positive attitude towards them. Developing students' ICT skills and instilling confidence in their ability to perform tasks involving these skills are further recommendations that can be drawn from the findings of this study. Educational institutions should take various measures to cultivate a sense of utility regarding m-learning, leading to improved performance, increased satisfaction, and a positive attitude towards m-learning. Following this observation, we have added an extended paragraph along the lines 662-677 in the revised manuscript.

Overall, "The Perception of Economical High School Students Toward M-Learning" is a valuable contribution to the literature. While there are some limitations to the study, they do not detract significantly from its overall quality. Therefore, I recommend a revision, which should include expanding the discussion parts and consulting some previous literature.

We appreciate the Referee’s favorable report and the enlightening criticisms.

Reviewer 2 Report

In the introduction, the initial statements about pedagogy should be justified with references, lines 38 to 45.
Idem lines 48-52; 65-70
The article is well founded and the methodological development is clear and coherent with the approach.

The contributions of the article are in line with other research carried out in other fields.
They are well structured but do not add a great novelty.

Author Response

Attached File: 

Response to Referee 2

We would like to thank the Referee for the concise report and constructive points she/he made. We are especially glad that the Referee finds our work “clear and coherent”. Below, we provide our responses to the report. Texts quoted from the referee reports are in black, while our responses are in blue. In the revised manuscript, all the modifications are indicated in red for easy identification.

In the introduction, the initial statements about pedagogy should be justified with references, lines 38 to 45. Idem lines 48-52; 65-70

Thank you for bringing this to our attention. In the revised version of the manuscript references related to the presented content have been added in the mentioned paragraphs, as visible in the revised version of the manuscript.

The article is well founded and the methodological development is clear and coherent with the approach. The contributions of the article are in line with other research carried out in other fields.
They are well structured but do not add a great novelty.

The main novelty of this work lies in examining the attitude of high school students studying economics towards m-learning. As mentioned in the manuscript, there is a gap in the research regarding the use of m-learning in economic education at the high school level, with very few studies addressing this topic. The increased interest is predominantly observed at the higher education level. Students studying economics are the ones who will be active participants in the future digital economy, and their training in this digitalized context is crucial for future economic development. To emphasize better the novelty of our work we have added paragraphs along the lines 152-157 and 622-631 in the revised manuscript.

Reviewer 3 Report

Firstly, the rationale of the study is stated in the introduction, however, the purpose/objective of the study needs to be mentioned point-wise for clarity.

the contextualization of the theoretical background, in regards to the previous and present times, barely mention to TAM, however, there isn't in depth description of if it being a new or old theory, as well as the fact that other theories are only mentioned and not described.

The research design is clearly stated as well as, the hypotheses, the method used, and the research questions. The argument, however, should be more coherent and balanced, there isn't a strong understanding of what the author's argument is,

the Author should focus on making it a lot more compelling.

The Author could have discussed the educator's perspective in depth as very little information from past research in regards is provided on the use of mobile applications

The results, however, are clearly stated, the positive relationship between perceived usefulness, perceived ease of use, and the attitude of students towards mobile devices in the learning environment. however, I suggest that the authors should elaborate on it.

The author should explain how much novelty of the objective is satisfied by the study conducted.

the quality of English is rated above average . it's quite commendable.

the language is clear and easily understandable.

Author Response

Attached File: 

Response to Referee 3

We would like to thank the Referee for the positive report and constructive points she/he made. Below, we provide our responses to the report. Texts quoted from the referee reports are in black, while our responses are in blue. In the revised manuscript, all the modifications are indicated in red for easy identification.

Firstly, the rationale of the study is stated in the introduction, however, the purpose/objective of the study needs to be mentioned point-wise for clarity.

We agree that in the original version of the manuscript, the main objectives were not that clearly presented. Therefore, to increase the readability we have provide further clarifications, additions have been made to lines 152-157 and 168-171 in order to provide a clearer explanation.

The contextualization of the theoretical background, in regards to the previous and present times, barely mention to TAM, however, there isn't in depth description of if it being a new or old theory, as well as the fact that other theories are only mentioned and not described.

We acknowledge that the TAM model is well-known, and we believe that the discussion we have presented in the manuscript provides a clear understanding of its application. The discussion related to the TAM model spans over one and a half pages, offering comprehensive insights. However, in response to the referee's suggestion, we have included examples illustrating how the TAM model has been employed to evaluate the attitude and behavioral intention of students and teachers regarding the use of mobile technologies.

The research design is clearly stated as well as, the hypotheses, the method used, and the research questions. The argument, however, should be more coherent and balanced, there isn't a strong understanding of what the author's argument is, the Author should focus on making it a lot more compelling. 

In order to enhance the clarity and significance of this work, we have made substantial revisions to both the Results and Discussion section and the Conclusions section. These modifications aim to better highlight the novelty of our findings and provide a more comprehensive perspective on the obtained results and their practical implications. We believe that these revisions greatly strengthen the overall message of the paper and make it more compelling to readers.

The Author could have discussed the educator's perspective in depth as very little information from past research in regards is provided on the use of mobile applications

This suggestion is highly valuable, and we will take it into consideration for future work. Given the complexity of the topic, it would indeed be challenging to address both students' and teachers' attitudes in the same study.

The results, however, are clearly stated, the positive relationship between perceived usefulness, perceived ease of use, and the attitude of students towards mobile devices in the learning environment. however, I suggest that the authors should elaborate on it. The author should explain how much novelty of the objective is satisfied by the study conducted.

In the revised version of the manuscript, we have expanded upon this point by enhancing the presentation in the Results and Discussion section. We hope that these revisions meet the expectations of the Referee. Moreover, in the conclusions section (lines 622-631, 641-652) in the revised manuscript we provide clarifications regarding the novelty of our study.

Reviewer 4 Report

Dear Authors,

Thank you for providing this article on the perception of economical high school students toward m-learning.  I find the topic of this study to be highly intriguing. Below I present my main concerns with the current form and some suggestions to improve it.

Title

1. As the objective of your study is to investigate the determinants of attitudes towards the use of mobile devices, as well as it is well illustrated in your research model (Figure 1), I suggest modifying the title of this paper as follows: "Determinants of economical high school students' attitudes toward mobile devices use".

Introduction

2. In the introductory section, the authors are asked to clarify the significance of the current study and outline the research gaps.

3. The introduction section has chunks of text without any single reference. For instance, in lines 48-52, "The COVID-19 pandemic has underscored the importance of flexible and intelligent educational systems. Moreover, the pandemic has also demonstrated that mobile learning has immense potential, and education systems worldwide are increasingly exploring, promoting, and embracing its new features. This trend indicates the sustainability of m-learning as an educational system", there is no in-text citation to support the claims. Kindly provide references to earlier studies in support of your statements.

4. The same remark for the paragraph from line 65 to line 71, " There is a wide range of reasons why adopting m-learning techniques is important, […], at times and places chosen by themselves". Similarly, for the lines 87-94, "On the contrary, certain studies have cast doubt on the actual advantages of mobile learning and its limitations. […], while ensuring that students do not receive assistance from others".

Literature Review on Mobile Learning

5. I recommend renaming this section as "Literature Review and Research Model" and suggest revising its structure as follows.

"2. Literature Review and Research Model

  2.1. Literature Review on Mobile Learning

 2.2. Research Model and the Hypotheses"

Methodology and Data Collection

6. In this section, a complete description of how data were acquired (data collection technique, data collection period, etc.) should be included.

7. Given that the data was collected online, it would be valuable to know if the authors conducted a pre-test of the questionnaire to ensure that it was clear and easily understandable to the participants. Therefore, it would be beneficial for the authors to provide details on the pre-testing process and any measures taken to ensure the questionnaire's comprehensibility.

8. The authors are requested to provide further clarification regarding the sampling method applied in this study.

9. In lines 421-422, the authors indicated, "The data was collected by a multiple item online survey. The questionnaire contains 24 items related to 7 constructs". However, upon reviewing the questionnaire provided in the appendix, it appears that there are only 23 items included. It would be helpful for the authors to clarify this discrepancy and provide an explanation for the discrepancy between the stated number of items and the actual number present in the questionnaire.

10. The authors are requested to provide a detailed explanation of the data analysis approach they used.

11. The paragraph from line 433 to line 437 illustrates outcomes related to measurement model assessment. This part should be included in the results section.

Results and Discussions

12. To test a hypothesis, it is not enough to present the β values, but it is necessary to check the level of significance of the relationship. Therefore, from line 471 to line 481, the authors are advised to include the values related to the level of significance of the relationships between the variables (P-value and t-value). These values should also be included in Figure 2.

13. I suggest that the authors should provide a more detailed analysis and interpretation of their study's findings, rather than simply reporting them in the results and discussion section. While it is important to report the study findings, it is equally important to help readers understand what these findings means in the context of the research questions and broader literature.  

Conclusion

14. To enhance the manuscript, I invite the authors to expand the conclusion section to provide more detailed theoretical and managerial implications of their study's findings.

Minor issues:

15. A number of sentences need to be rephrased because they are copied from other works without giving any credit to the original source. For instance, from line 471 to line 475 "In Figure 2 we present the main result of our analysis and give the path coefficients of the model which indicates that the hypotheses that we proposed and the associations between the constructs are valid. First, our findings validate the underlying TAM model in which the most significant impact on the ATMDU comes from the perceived usefulness (β≈0.642), followed by perceived ease of use (β≈0.190)", copied from https://doi.org/10.1007/978-3-031-09421-7 . Please reduce the index of similarity. If it is about one of your earlier works, kindly avoid self-plagiarism.

I hope these comments are useful for further steps.

All the best

The paper is easy to follow, but there are some grammar mistakes if you look at it carefully.

Author Response

Attached File: 

Response to Referee 4

We would like to thank the Referee for the constructive points she/he made. We express our gratitude for the Referee's thorough report. Below, we provide our responses to the report. Texts quoted from the referee reports are in black, while our responses are in blue. In the revised manuscript, all the modifications are indicated in red for easy identification.

Thank you for providing this article on the perception of economical high school students toward m-learning. I find the topic of this study to be highly intriguing. Below I present my main concerns with the current form and some suggestions to improve it.

Title

1. As the objective of your study is to investigate the determinants of attitudes towards the use of mobile devices, as well as it is well illustrated in your research model (Figure 1), I suggest modifying the title of this paper as follows: "Determinants of economical high school students' attitudes toward mobile devices use".

We appreciate your suggestion, and upon considering the informative title proposed by the Referee, we have decided to revise the title accordingly.

Introduction

2. In the introductory section, the authors are asked to clarify the significance of the current study and outline the research gaps.

Referee 1 also raised this point, prompting us to enhance the introduction section by incorporating a new paragraph that delves into the m-learning situation during the Covid-19 period. Additionally, we clarified various aspects regarding the significance of our study, particularly its focus on the limited number of studies conducted on this topic at the secondary school level. Previous research predominantly concentrated on higher education. Furthermore, in response to the Referee's request, we have provided a clearer exposition of the main objectives of our research. These modifications have been integrated into the revised manuscript between lines 152 and 157, as well as lines 168 and 171.

3. The introduction section has chunks of text without any single reference. For instance, in lines 48-52, "The COVID-19 pandemic has underscored the importance of flexible and intelligent educational systems. Moreover, the pandemic has also demonstrated that mobile learning has immense potential, and education systems worldwide are increasingly exploring, promoting, and embracing its new features. This trend indicates the sustainability of m-learning as an educational system", there is no in-text citation to support the claims. Kindly provide references to earlier studies in support of your statements.

The absence of references in this paragraph was also highlighted by the second referee. Consequently, we have addressed this concern by including a list of references to support the content.

4. The same remark for the paragraph from line 65 to line 71, "There is a wide range of reasons why adopting m-learning techniques is important, […], at times and places chosen by themselves". Similarly, for the lines 87-94, "On the contrary, certain studies have cast doubt on the actual advantages of mobile learning and its limitations. […], while ensuring that students do not receive assistance from others".

We have fixed this issue as well by adding new references to support the message.

Literature Review on Mobile Learning

5. I recommend renaming this section as "Literature Review and Research Model" and suggest revising its structure as follows.

"2. Literature Review and Research Model

 2.1. Literature Review on Mobile Learning

 2.2. Research Model and the Hypotheses"

While the Referee's suggestion is indeed useful, our main concern revolves around the potential proliferation of sub-sections within an already existing subsection. This has the potential to adversely impact the overall structure and readability of the paper. Therefore, after careful consideration, we have decided to retain the standard structure that is commonly preferred in most of the papers.

Methodology and Data Collection

6. In this section, a complete description of how data were acquired (data collection technique, data collection period, etc.) should be included.

In the revised manuscript, we have modified the title of Section 4 to "Methodology, Data Collection, and Data Analysis." Additionally, we have included a new paragraph that provides a detailed description of the data collection process. In the revised manuscript, this particular section can be found between lines 459 and 478.

7. Given that the data was collected online, it would be valuable to know if the authors conducted a pre-test of the questionnaire to ensure that it was clear and easily understandable to the participants. Therefore, it would be beneficial for the authors to provide details on the pre-testing process and any measures taken to ensure the questionnaire's comprehensibility.

To assess the questionnaire's quality and comprehensibility, a pre-test was initially conducted with a class of students. Any ambiguities identified in the statements were addressed, and reformulations were made to enhance the level of understanding. Furthermore, Google Forms, which was used as the survey platform, underwent a critical evaluation and underwent some formatting modifications as well. To address this point raised by the Referee a paragraph discussing these aspects has been included in the revised manuscript, specifically located at lines 468-472.

8. The authors are requested to provide further clarification regarding the sampling method applied in this study.

The sampling was carefully designed to be representative of the entire student population included in this study. Both male and female students from all levels of study were considered, as shown in Table 1, across all four profiles: economy, administration, trade, and tourism. We provide more details in the revised manuscript by including a paragraph about the sampling located at the lines 445-457

9. In lines 421-422, the authors indicated, "The data was collected by a multiple item online survey. The questionnaire contains 24 items related to 7 constructs". However, upon reviewing the questionnaire provided in the appendix, it appears that there are only 23 items included. It would be helpful for the authors to clarify this discrepancy and provide an explanation for the discrepancy between the stated number of items and the actual number present in the questionnaire.

A typo was in the manuscript, and the accurate number of items is 23. This error has been rectified in the revised version.

10. The authors are requested to provide a detailed explanation of the data analysis approach they used.

In accordance with the Referee's request, we have incorporated a new paragraph in the revised manuscript that provides a explanation of the data analysis process. Moreover, in line with item 6 of this response letter, the title of the section has been appropriately modified to reflect these changes. The paragraph related to this point is in between the lines 479-489.

11. The paragraph from line 433 to line 437 illustrates outcomes related to measurement model assessment. This part should be included in the results section.

This is a valid point, and we have addressed it by rearranging the corresponding tables and placing them in their correct positions.

Results and Discussions

12. To test a hypothesis, it is not enough to present the β values, but it is necessary to check the level of significance of the relationship. Therefore, from line 471 to line 481, the authors are advised to include the values related to the level of significance of the relationships between the variables (P-value and t-value). These values should also be included in Figure 2.

In response to this feedback, we have made the necessary revisions in the manuscript. We have included the p-values and t-values in the text to provide a comprehensive analysis. Although we initially considered including figures with these values, upon further consideration, we concluded that the figures may contain excessive numbers, potentially leading to confusion. Hence, we have chosen to retain the p and t-values in the text for clarity and accuracy but we have added an extra Table 5 where we present the results suggested by the Referee.  

13. I suggest that the authors should provide a more detailed analysis and interpretation of their study's findings, rather than simply reporting them in the results and discussion section. While it is important to report the study findings, it is equally important to help readers understand what these findings means in the context of the research questions and broader literature.

This point was raised by Referee 1 as well, highlighting the schematic nature of the results discussion, which was initially limited for the sake of manuscript length. In response, we have expanded this section in the revised manuscript by incorporating more in-depth discussions and establishing connections with other related works. Additionally, we have included paragraphs specifically addressing the practical implications of our study. In the revised manuscript, these paragraphs can be found between lines 537 and 596.

Conclusion

14. To enhance the manuscript, I invite the authors to expand the conclusion section to provide more detailed theoretical and managerial implications of their study's findings.

In the original manuscript, the conclusion was somewhat concise. However, in response to the Referee's suggestion, we have expanded it to delve into more comprehensive discussions regarding the theoretical and practical implications of our study. In the revised manuscript, this extended conclusion can be found in the following sections: between lines 622 and 631, 641 and 652, as well as lines 662 and 677.

Minor issues:

15. A number of sentences need to be rephrased because they are copied from other works without giving any credit to the original source. For instance, from line 471 to line 475 "In Figure 2we present the main result of our analysis and give the path coefficients of the model which indicates that the hypotheses that we proposed and the associations between the constructs are valid. First, our findings validate the underlying TAM model in which the most significant impact on the ATMDU comes from the perceived usefulness (β≈0.642), followed by perceived ease of use (β≈0.190)", copied from https://doi.org/10.1007/978-3-031-09421-7. Please reduce the index of similarity. If it is about one of your earlier works, kindly avoid self-plagiarism.

We have changed the paragraph to avoid such an issue. Thank you for pointing it out.  

I hope these comments are useful for further steps.

Indeed, the comments provided were highly valuable to us, as we firmly believe they will significantly enhance the readability of our manuscript.

Round 2

Reviewer 4 Report

Dear Authors,

I am delighted to inform you that your manuscript has successfully addressed all the recommended revisions from the previous review. Based on the improvements made, I am confident in recommending it for publication.

Minor editing of English language required.